# Curcumin Attenuates Liver Steatosis via Antioxidant and Anti-Inflammatory Pathways in Obese Patients with Type 2 Diabetes Mellitus: A Randomized Controlled Trial

**DOI:** 10.3390/ijms26199286

**Published:** 2025-09-23

**Authors:** Metha Yaikwawong, Khanittha Kamdee, Somlak Chuengsamarn

**Affiliations:** 1Department of Pharmacology, Faculty of Medicine Siriraj Hospital, Mahidol University, Bangkok 10700, Thailand; metha.yai@mahidol.ac.th (M.Y.); khanittha.kam@mahidol.ac.th (K.K.); 2Division of Endocrinology and Metabolism, Faculty of Medicine, HRH Princess Maha Chakri Sirindhorn Medical Center, Srinakharinwirot University, Nakhon Nayok 26120, Thailand

**Keywords:** type 2 diabetes, curcumin, nonalcoholic fatty liver disease, metabolic dysfunction-associated steatotic liver disease, nutritional supplements

## Abstract

Liver steatosis, the hallmark component of metabolic dysfunction-associated steatotic liver disease (MASLD), is particularly common among individuals with type 2 diabetes mellitus (T2DM). Shared mechanisms such as insulin resistance, oxidative stress, and chronic inflammation contribute to the coexistence of these conditions and accelerate disease progression, emphasizing the need for effective therapeutic strategies. In this 12-month, randomized, double-blind, placebo-controlled trial, 227 obese individuals with T2DM were assigned to receive either 1500 mg of curcumin daily or placebo. Curcumin significantly reduced liver fat content, liver stiffness, and glycated hemoglobin (HbA1c) compared with placebo (all *p* < 0.001). Improvements were also noted in inflammatory mediators, including interleukin-1 beta (IL-1β) and tumor necrosis factor-alpha (TNF-α) (all *p* < 0.001), reflecting curcumin’s anti-inflammatory effects. Antioxidant benefits were evident, as total antioxidant capacity (TAC), glutathione peroxidase (GPx), and superoxide dismutase (SOD) increased, while malondialdehyde levels decreased (all *p* < 0.001). Systematic safety assessments, including liver and kidney function tests, revealed no clinically significant abnormalities. Mild gastrointestinal discomfort was the most common non-serious adverse event. Overall, these findings support curcumin as a safe and effective adjunctive therapy for improving liver steatosis in obese patients with T2DM.

## 1. Introduction

The global prevalence of type 2 diabetes mellitus (T2DM) continues to rise at an alarming pace. According to the International Diabetes Federation, 10.5% of adults worldwide are currently affected, and this figure may reach one in eight adults by 2045 if current trends persist [1]. In parallel, metabolic dysfunction-associated steatotic liver disease (MASLD)—recently redefined from non-alcoholic fatty liver disease (NAFLD)—has become a major public health concern, affecting up to 38% of the global population [2].

MASLD and T2DM are tightly interconnected through shared mechanisms—insulin resistance, metabolic dysregulation, and chronic inflammation. MASLD predisposes to T2DM, while T2DM accelerates progression to advanced liver [3,4,5].

Oxidative stress and inflammation play central roles in MASLD pathogenesis. Hepatic triglyceride accumulation promotes mitochondrial dysfunction and excessive production of reactive oxygen species (ROS), overwhelming endogenous antioxidant defenses such as superoxide dismutase (SOD), catalase, and glutathione peroxidase (GPx) [6,7,8,9]. The resulting imbalance contributes to hepatocellular injury, inflammation, and fibrosis.

Curcumin, the principal polyphenol in Curcuma longa (turmeric), has attracted attention for its hepatoprotective potential. Preclinical studies demonstrate antioxidant, anti-inflammatory, antifibrotic, and lipid-modulating effects [10,11,12,13,14,15]. Clinical trials in patients with NAFLD/MASLD further support these findings, reporting improvements in steatosis, liver enzymes, and metabolic parameters. For example, a 72-week study of phospholipid curcumin demonstrated regression of significant liver fibrosis, improved glycemic control, favorable modulation of lipid profiles, and reduced systemic inflammation in patients with nonalcoholic steatohepatitis [16]. Similarly, a 24-week double-blind randomized trial in 80 patients with nonalcoholic simple fatty liver showed that curcumin supplementation (500 mg/day) significantly reduced hepatic fat content, body weight, and BMI, with additional benefits on free fatty acids, triglycerides, fasting blood glucose, HbA1c, and insulin levels [17]. A 12-week randomized controlled trial of phytosomal curcumin in NAFLD patients also reported significant reductions in hepatic steatosis, fibrosis, and waist circumference compared with placebo [18]. The intervention significantly reduced hepatic steatosis and fibrosis compared to placebo. Despite these encouraging results, most studies have been limited by small sample sizes and short intervention durations (8–24 weeks), leaving the long-term efficacy and safety of curcumin supplementation in high-risk T2DM populations uncertain.

Beyond curcumin, diarylheptanoids—a structurally related class of polyphenolic compounds—have also shown potent antioxidant and anti-inflammatory properties. Recent reviews highlight their ability to regulate oxidative stress pathways, inhibit pro-inflammatory cytokines, and improve metabolic function relevant to MASLD and T2DM [19]. These parallels emphasize the importance of structure-activity relationships and support the translational rationale for curcumin as part of a broader class of phytochemicals with therapeutic potential in hepatic steatosis and metabolic inflammation.

In this context, we conducted a 12-month, randomized, double-blind, placebo-controlled trial to evaluate the effects of curcumin supplementation in obese patients with T2DM. The primary objective was to determine whether curcumin could reduce liver steatosis, assessed non-invasively using the controlled attenuation parameter (CAP). Secondary outcomes included changes in liver stiffness, antioxidant and inflammatory markers, lipid profiles, and glycemic parameters. By extending the intervention period and enrolling a larger cohort, this study addresses the critical knowledge gap regarding the long-term therapeutic potential and safety of curcumin in MASLD among patients with T2DM.

## 2. Results

Figure 1 illustrates the trial flow chart, and Table 1 presents the baseline characteristics of the 227 randomized participants. No significant differences were observed between groups at baseline, ensuring comparability. Nutrient intake was assessed at baseline and after 12 months, as shown in Table 2. Analysis of total energy intake, macronutrient distribution, and fiber intake revealed no significant intergroup differences at either time point. This consistency in dietary patterns throughout the study period minimizes the likelihood of dietary confounding and supports the validity of the intervention’s outcomes.

### 2.1. Intervention Outcomes

#### 2.1.1. Curcumin Treatment Improved Liver Steatosis and Liver Stiffness

Curcumin treatment significantly reduced both liver steatosis and liver stiffness throughout the 12-month study. A statistically significant difference was observed for both outcomes at the 3, 6, 9, and 12-month time points when compared to the placebo group (Table 2; Figure 2A).

#### 2.1.2. Antioxidant Effects

Compared to the placebo group, curcumin supplementation significantly enhanced antioxidant defenses, as evidenced by increased levels of total antioxidant capacity (TAC), glutathione peroxidase (GPx), and superoxide dismutase (SOD) activity at 3, 6, 9, and 12 months. In parallel, curcumin treatment led to a significant reduction in malondialdehyde (MDA) levels—a key biomarker of oxidative stress—across the same time points (Table 3).

#### 2.1.3. Anti-Inflammatory Effects

Curcumin significantly reduced the pro-inflammatory biomarkers IL-1β and TNF-α, (Table 3).

#### 2.1.4. Anthropometric Measurement Effect

Anthropometric measurements, including BMI and WC, were significantly lower in the curcumin group compared to the placebo group at 3, 6, 9, and 12 months (Table 4; Figure 2B).

#### 2.1.5. Glycemic Control Effect

Compared with placebo, curcumin treatment significantly reduced FPG and HbA1c (key markers of diabetes progression) at 6, 9, and 12 months (Table 4; Figure 2C).

#### 2.1.6. Insulin Sensitivity Index

Compared with placebo, curcumin treatment significantly increased QUICKI at 3, 6, 9, and 12 months (Table 4).

#### 2.1.7. Insulin Resistance Index

Compared with placebo, curcumin treatment significantly reduced TyG-WC—non-invasive markers of insulin resistance and are closely associated with MASLD—at 3, 6, 9, and 12 months (Table 4).

#### 2.1.8. Lipid Profiles

Curcumin supplementation resulted in a significant reduction in total cholesterol and LDL-C levels when compared to the placebo group at the 6, 9, and 12-month time points. Furthermore, TG and the TyG-WC index, a measure of insulin resistance, were significantly lowered at the 3, 6, 9, and 12-month intervals (Table 5).

#### 2.1.9. Non-Esterified Fatty Acid Levels

Curcumin supplementation significantly decreased NEFA levels at 6 months compared to placebo (Table 5). Given NEFA’s role in promoting hepatic fat accumulation and contributing to the pathogenesis of MASLD, this reduction suggests improved hepatic metabolic function and attenuated lipotoxicity.

#### 2.1.10. Adverse Effects

Reported mild adverse events included abdominal discomfort, loose stools, and headaches. No participants withdrew from the study due to these effects (Table 6). To evaluate the safety of curcumin, kidney and liver function were monitored using established biomarkers (Table 7). A comparison of aspartate transaminase, alanine transaminase, and creatinine levels revealed no significant differences between the curcumin and placebo groups. Furthermore, no instances of hypoglycemia were recorded in the curcumin group. Overall, these results support the long-term safety of the curcumin extract for up to 12 months. Comparable capsule consumption rates across both groups (Table 3) indicate similar levels of compliance, thus ruling out differential adherence as a confounding factor for the study’s outcomes.

## 3. Discussion

Liver steatosis, defined as fat accumulation in more than 5% of hepatocytes, represents the hallmark of MASLD and is particularly prevalent among individuals with type T2DM [4]. The bidirectional relationship between MASLD and T2DM is well established: MASLD predisposes to incident T2DM, while T2DM accelerates progression to advanced liver disease, including fibrosis, cirrhosis, and hepatocellular carcinoma [20,21]. Shared mechanisms—most notably insulin resistance, lipotoxicity, oxidative stress, and chronic low-grade inflammation—underlie this interplay. Targeting these converging pathways therefore remains essential in developing effective therapeutic strategies [3,22].

Curcumin, the bioactive compound in turmeric, exhibits potent antioxidant and anti-inflammatory properties that may be beneficial in managing liver diseases and type 2 diabetes mellitus (T2DM). These effects help mitigate liver inflammation and may slow the progression of liver diseases like MASLD [23,24].

Although liver biopsy remains the gold standard for MASLD diagnosis and staging, its invasiveness and limited feasibility in large trials restrict its routine use. MRI-proton density fat fraction (MRI-PDFF) provides a highly accurate non-invasive alternative but is costly and not widely accessible [25]. For feasibility, we employed controlled attenuation parameter (CAP) as a validated non-invasive surrogate, suitable for longitudinal monitoring in large cohorts [26].

Over the course of a 12-month randomized controlled trial, curcumin supplementation significantly improved hepatic steatosis, liver stiffness, oxidative stress, inflammatory cytokines, lipid parameters, glycemic indices, and insulin resistance in patients with T2DM and obesity. These findings support curcumin’s pleiotropic properties and alignment with MASLD pathophysiology. However, while statistically significant, not all improvements are clinically meaningful. The average CAP reduction (−38.60 dB/m) likely reflects a true decrease in hepatic fat [27]. By contrast, the changes in antioxidant enzymes or lipid fractions may represent mechanistic activity rather than direct clinical benefit.

The antioxidant effects of curcumin were demonstrated by increases in TAC, GPx, and SOD, together with reductions in MDA. These findings indicate reinforcement of endogenous antioxidant defenses and attenuation of lipid peroxidation. Curcumin’s well-documented ability to scavenge reactive oxygen species and activate the Nrf2 signaling pathway likely underpins these benefits [28,29,30,31]. Since oxidative stress drives hepatocellular injury and the transition from steatosis to steatohepatitis, curcumin’s capacity to mitigate oxidative damage may help stabilize mitochondrial function and reduce hepatic lipid accumulation [32,33].

Anti-inflammatory effects were also evident, with significant reductions in IL-1β and TNF-α. Notably, improvements in hepatic steatosis occurred synchronously with these cytokine reductions, suggesting that anti-inflammatory activity is a core mechanism by which curcumin alleviates hepatic fat accumulation. Chronic low-grade inflammation links insulin resistance and MASLD, and suppression of NF-κB may interrupt this cycle of hepatocellular injury, immune activation, and fibrogenesis [34,35,36].

Curcumin further improved metabolic parameters, including triglycerides, LDL-C, NEFA, BMI, and WC, alongside better insulin sensitivity indices (QUICKI, TyG-WC). The transient NEFA rise at 3 months followed by sustained reductions suggests a biphasic response—initial fatty acid mobilization, followed by improved insulin sensitivity and fatty acid oxidation [37].

A reduction in liver stiffness was also observed. Given the short study duration, this is more likely attributable to decreased hepatic inflammation rather than true fibrosis regression, but longer-term use may yield structural benefits [38].

Curcumin at 1500 mg/day was well tolerated, with only mild gastrointestinal adverse effects reported, consistent with its established safety profile. Nevertheless, limited bioavailability and potential herb-drug interactions remain important considerations, especially in polypharmacy among T2DM patients [39,40].

The strengths of this study include its extended duration, relatively large sample size, and comprehensive assessment of oxidative stress, inflammatory, and metabolic parameters. Limitations include the reliance on CAP rather than liver biopsy or MRI-PDFF [41] for assessing hepatic steatosis, the single-center Thai cohort, and testing of a single curcumin dose without pharmacokinetic monitoring. Additionally, restricting enrollment to patients on metformin monotherapy with well-controlled glycemia minimized confounding but reduced generalizability to real-world MASLD-T2DM cohorts, which often involve severe metabolic dysfunction and diverse treatments. Future studies should evaluate curcumin in broader populations and directly compare it with emerging MASLD therapies such as GLP-1 receptor agonists and SGLT2 inhibitors [42,43].

## 4. Materials and Methods

### 4.1. Study Design and Participants

This randomized, double-blind, placebo-controlled trial was conducted at the HRH Princess Maha Chakri Sirindhorn Medical Center of Srinakharinwirot University, Nakhon Nayok, Thailand. A total of 227 patients with T2DM were selected for enrollment based on predefined inclusion and exclusion criteria (see Figure 3 for a complete flow chart). The 12-month study included a 3-month run-in period prior to randomization, during which all participants received standardized education on diet and exercise protocols. Written lifestyle recommendations and a 20–30 min one-on-one workshop emphasizing the importance of a healthy lifestyle were provided. Participants were encouraged to adhere to medical nutrition therapy guidelines and engage in regular physical activity.

To minimize interference from other antidiabetic medications, only T2DM patients using metformin for glycemic control were recruited. Participants with hypertension and dyslipidemia were maintained on existing antihypertensive and antidyslipidemic regimens, with no adjustments allowed during the study. The inclusion criteria were age ≥35 years, T2DM diagnosis within the past year, well-controlled blood glucose (glycated hemoglobin [HbA1c] <6.5% and fasting plasma glucose <110 mg/dL), and body mass index ≥23 kg/m^2^.

The diagnosis of T2DM was established in accordance with the 2017 guidelines issued by the American Diabetes Association [45]. Eligibility for inclusion required meeting at least one of the following diagnostic thresholds: fasting plasma glucose ≥126 mg/dL, 2 h plasma glucose ≥200 mg/dL following an oral glucose tolerance test, HbA1c ≥6.5%, or random plasma glucose ≥200 mg/dL in conjunction with typical symptoms of hyperglycemia or a hyperglycemic emergency. Ethical clearance for the study was granted by the Ethics Committee of the Faculty of Medicine, Srinakharinwirot University, Bangkok, Thailand (approval number MEDSWUEC-113/2555), and the trial was registered with the Thai Clinical Trials Registry (ID: 20140303003). All procedures adhered to the principles outlined in the Declaration of Helsinki, and written informed consent was obtained from all participants prior to enrollment.

Individuals diagnosed with type 1 diabetes, gestational diabetes, impaired glucose tolerance, or maturity-onset diabetes of the young were excluded from participation. Blood samples were collected after overnight fasting at baseline and subsequently at 3-month intervals up to 12 months. Participants with an HbA1c ≥7∙0% or a fasting plasma glucose ≥130 mg/dl on two consecutive tests during the intervention period were excluded (Figure 1). Patients with uncontrolled hypertension (two readings ≥140/90 mmHg) or dyslipidemia (two low-density lipoprotein cholesterol [LDL-C] readings ≥130 mg/dL) were also excluded. Details on specific medications are provided in Table 8.

Prior to the study, all subjects received dietary and exercise counseling. A 3-day food intake record (2 weekdays, 1 weekend day) was analyzed using Computer Dietary Guidance System Software version 3.0 (CDGSS 3∙0) software to estimate nutrient intake at baseline and 12 weeks. A questionnaire was used at baseline to assess dietary habits, including the frequency and quantity of meat, milk, egg, and vegetable consumption (Table 9).

### 4.2. Randomization Procedures

After screening and obtaining informed consent, participants received guidance on diet and lifestyle before being randomly assigned using a computer-generated sequence or random number tables. To minimize selection bias, the allocation sequence was concealed from staff involved in participant enrollment. Eligible individuals were then assigned to either the treatment or control arm based on this concealed randomization protocol. The integrity of blinding procedures was routinely assessed, and consistency in the controlled attenuation parameter (CAP) measurements and liver stiffness were maintained throughout the study. All aspects of the randomization process, including sequence generation and group allocation, were securely recorded for later analysis.

### 4.3. Blinding Procedures

To maintain double-blinding, placebo capsules were designed to closely resemble the curcumin capsules in appearance, taste, and method of administration. All participants received their assigned interventions according to a uniform protocol, ensuring blinding was preserved throughout the study duration. Healthcare professionals followed standardized procedures to reduce potential bias and were trained to avoid revealing treatment details. Each treatment container was labeled with coded identifiers, accessible only to the study coordinator, ensuring allocation concealment was upheld.

### 4.4. Preparation of Curcuminoid Capsules

Turmeric rhizomes (*Curcuma longa* Linn.) harvested from Kanchanaburi Province, Thailand, were dried, pulverized, and extracted using ethanol. The solvent was removed under reduced pressure, resulting in a semisolid extract rich in curcuminoids and oleoresin. Subsequent separation of oleoresin yielded a curcuminoid-rich fraction containing approximately 75–85% total curcuminoids. The relative proportions of curcumin, demethoxycurcumin, and bisdemethoxycurcumin were quantified using high-performance thin-layer chromatography. The final product, standardized to contain 250 mg of curcuminoids per dose, was encapsulated following Good Manufacturing Practice (GMP) protocols. Chromatographic profiles and detailed chemical composition are illustrated in Figure 4.

The high-performance thin-layer chromatography (HPTLC) chromatogram of the Thai Government Pharmaceutical Organization (GPO) is shown in Figure 2, compared with HPTLC chromatogram of standard across curcumin (curcuminoids) in Figure 1. In every batch of GPO curcuminoid extracts, the peak ratio of curcumin to demethoxycurcumin to bisdemethoxycurcumin was controlled to be 1 to not more than 0.6 to not more than 0.4.

### 4.5. Intervention

Participants ingested three blinded capsules of either curcumin (250 mg of curcuminoids per capsule) or placebo twice daily (six capsules per day) for 12 months. Both the curcumin and placebo capsules were manufactured by the Government Pharmaceutical Organization of Thailand. Capsule compliance was assessed at the 3-, 6-, 9-, and 12-month follow-up visits, and capsule counts were recorded (Table 10).

### 4.6. Study Outcomes

The primary outcome of this study, liver steatosis, was noninvasively assessed using transient elastography (FibroScan 520 Touch; Echosens, Paris, France). The controlled attenuation parameter (CAP) was measured at enrollment following a minimum 6 h fast. All scans were conducted by a single trained operator with extensive experience ( >200 determinations) who used the M probe (or the XL probe when necessary) to ensure at least 10 valid measurements were obtained with an interquartile range (IQR) of less than 0.2.

Secondary outcomes included changes in liver stiffness, total antioxidant capacity, superoxide dismutase, glutathione peroxidase, malondialdehyde, total cholesterol, triglycerides, non-esterified fatty acid, fasting plasma glucose, HbA1c, and insulin sensitivity/resistance indices (quantitative insulin sensitivity check index [QUICKI] and triglyceride-glucose index with waist circumference [TyG-WC]), interleukin-1 beta (IL-1β), and tumor necrosis factor-alpha (TNF-α). Adverse effects were defined as elevated creatinine (≥1.2 mg/dL), aspartate transaminase/alanine transaminase elevation (≥3× the upper limit of normal), and any reported symptoms [46].

### 4.7. Data Collection and Measurement Methods

Measurements were conducted at baseline and at 3-, 6-, 9-, and 12-months post-intervention. Baseline assessments included demographic data, medical history, medication use, body weight, height, waist circumference, and vital signs. Waist circumference, an indicator of abdominal obesity, was measured horizontally midway between the iliac crest and the costal margin [47]. Body mass index (BMI) was assessed using a bioelectrical impedance analyzer (Omron HBF-362; Omron Healthcare Singapore Pte Ltd., Alexandra Technopark, Singapore) [48].

Fasting blood samples were collected from the antecubital vein at 8:00 AM following an overnight fast. These samples were used to evaluate cardiometabolic risk parameters, including fasting plasma glucose (FPG) and glycated hemoglobin (HbA1c).

Lipid profiles—total cholesterol, triglycerides, low-density lipoprotein cholesterol (LDL-C), and high-density lipoprotein cholesterol (HDL-C)—were measured using diagnostic kits from Randox Laboratories Ltd. (Antrim, UK) and analyzed with an automated analyzer (Schiapparelli Biosystems Inc, Columbia, MD, USA).

Insulin sensitivity and resistance were assessed using two indices: the quantitative insulin sensitivity check index (QUICKI) and the triglyceride-glucose index with waist circumference (TyG-WC).

QUICKI, a validated surrogate marker derived from fasting plasma glucose and insulin concentrations, was calculated using the following formula [49].$$QUICKI=\frac{1}{log\left(INS\right)+log(FPG)}$$
where INS = Fasting Insulin  [μU/mL]), FPG is fasting plasma glucose level (mg/dL),

The TyG-WC index, a composite marker of insulin resistance and central obesity, was calculated using the formula: [50]:$$TyG-WC=ln\frac{TG\times FPG}{2}\times WC$$
where, TG is triglyceride (mg/dL), FPG is fasting plasma glucose (mg/dL), WC is waist circumference (cm) and ln is natural logarithm.

Oxidative stress and antioxidant capacity were evaluated using a panel of biochemical markers. Serum malondialdehyde (MDA), indicative of lipid peroxidation, was quantified via the thiobarbituric acid reactive substances (TBARS) method. This involved incubating samples with thiobarbituric acid under acidic and high-temperature conditions to form MDA-TBA complexes, which were measured spectrophotometrically at 532 nm. Fluorescence detection was also performed at 547 nm (excitation at 525 nm) using a Kontron SFM 25A spectrofluorometer (Kontron, Milan, Italy) [51]. Total antioxidant capacity was assessed using an automated assay developed by Erel, which measures the ability of plasma to counteract hydroxyl radical-induced oxidative reactions [52]. Enzymatic antioxidant activities, including superoxide dismutase (SOD) and glutathione peroxidase (GPx), were determined using colorimetric assays with RANSOD and RANSEL kits (Randox Laboratories Ltd., Crumlin, UK) on an Abbott Alcyon 300 analyzer (Abbott Laboratories, Abbott Park, IL, USA). Serum levels of inflammatory cytokines—interleukin-1β (IL-1β) and tumor necrosis factor-alpha (TNF-α)—were measured using validated ELISA kits (Abcam, San Francisco, CA, USA; catalog numbers ab214025 and ab181421). All blood samples were processed under standardized laboratory conditions to maintain sample quality and reliability.

### 4.8. Sample Size

This study’s sample size was calculated to detect a statistically significant difference in controlled attenuation parameter (CAP) between the groups. Using a two-sided independent samples *t*-test, with a significance level (α) of 0.05 and 80% power, a sample size of 64 patients per group was determined to be sufficient to detect a mean difference of 30 dB/m in CAP, assuming a standard deviation of 45 dB/m [53].

### 4.9. Statistical Analysis

Baseline demographic characteristics were presented as medians with interquartile ranges for continuous variables, and as counts with percentages for categorical data. Outcome measures were reported as medians (IQR) at baseline and at 3-month intervals up to 12 months. A per-protocol approach was used for all outcome analyses. Group comparisons were performed using independent *t*-tests for data with normal distribution and Mann-Whitney U tests for non-normally distributed variables. The Shapiro-Wilk test was applied to assess data normality. Categorical variables were analyzed using either chi-square or Fisher’s exact tests, depending on distribution and sample size. All statistical procedures were conducted using R software, version 4.3.2.

## 5. Conclusions

Curcumin, a bioactive polyphenol derived from turmeric, exhibits hepatoprotective, antioxidant, and anti-inflammatory properties. In this 12-month randomized controlled trial, curcumin supplementation improved hepatic steatosis, glycemic control, lipid profiles, and markers of oxidative stress and inflammation in obese patients with T2DM. These findings support curcumin as a promising adjunctive therapy for MASLD. Nevertheless, confirmation of its therapeutic role requires multicenter trials with varied dosing regimens, longer follow-up, and direct comparisons with established pharmacotherapies.

## Figures and Tables

**Figure 1 ijms-26-09286-f001:**
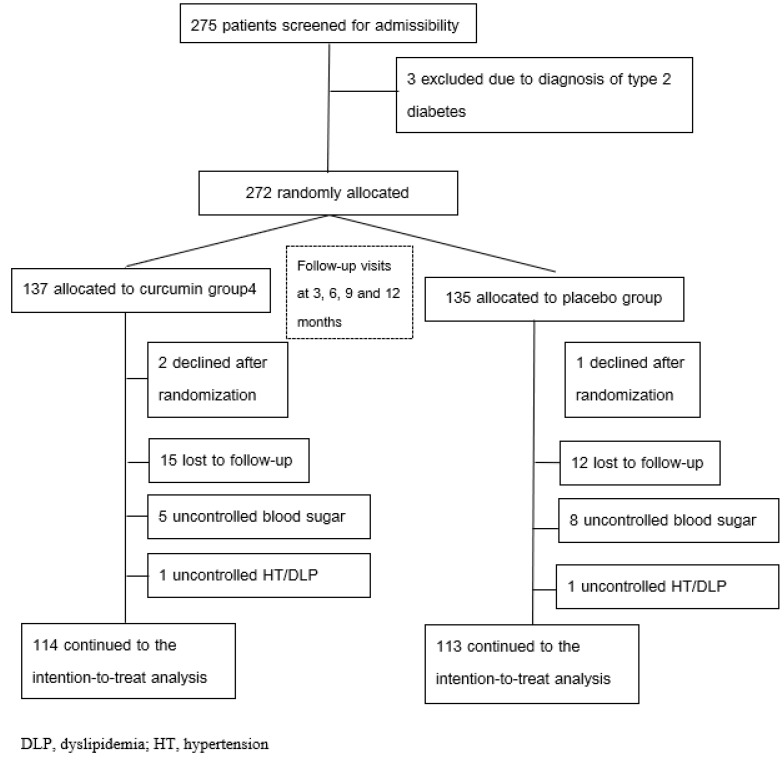
Trial profile (CONSORT Diagram).

**Figure 2 ijms-26-09286-f002:**
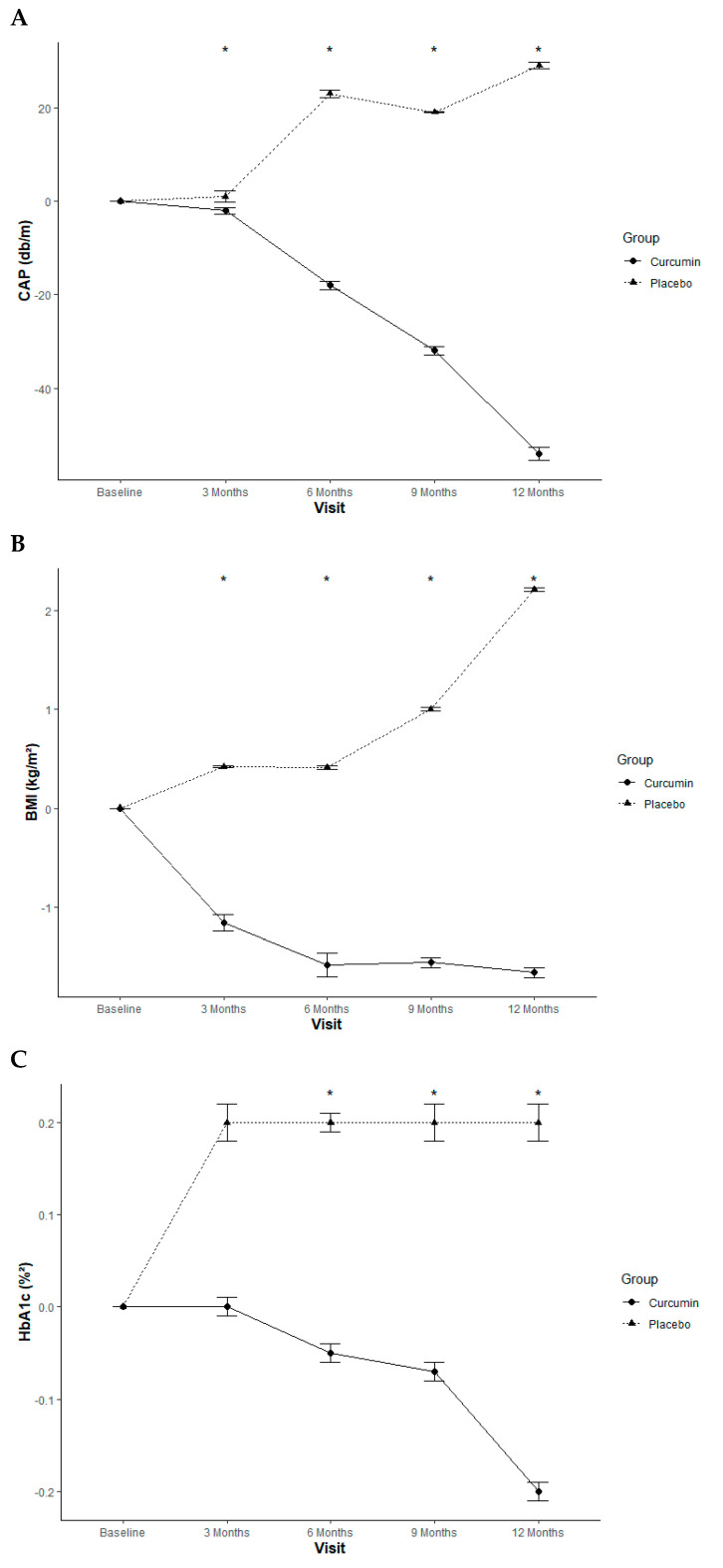
Mean of parameters with SEM at baseline, 3, 6, 9 and 12 months were compared between placebo- and curcumin-treated group. (**A**) Controlled Attenuation pressure (CAP); (**B**) Body mass index (BMI); (**C**) Glycated hemoglobin (HbA1c) * Statistically significant.

**Figure 3 ijms-26-09286-f003:**
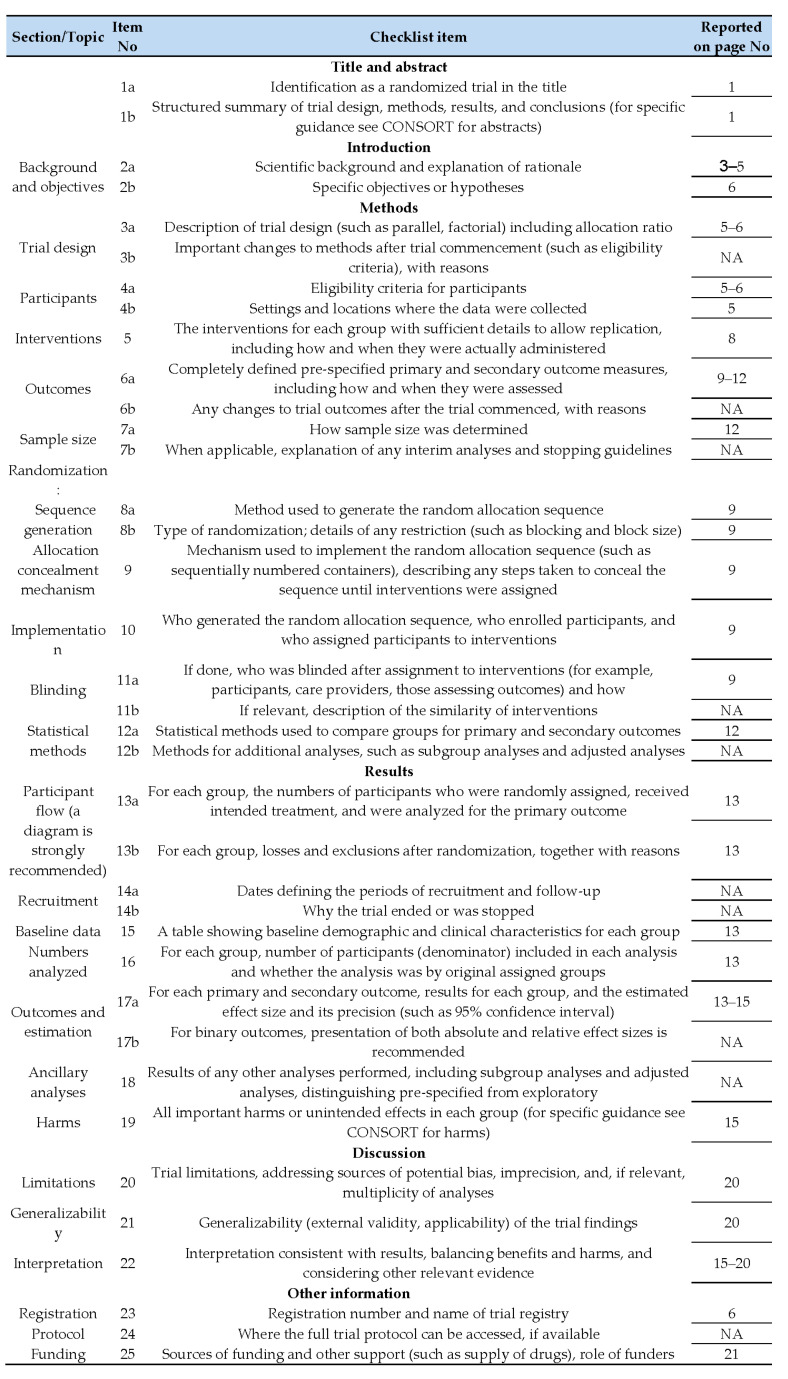
CONSORT 2010 checklist for reporting a randomized trial. Citation: Schulz KF, Altman DG, Moher D, for the CONSORT Group. CONSORT 2010 Statement: updated guidelines for reporting parallel group randomized trials. BMC Medicine. 2010;8:18. © 2010 [44] Schulz et al. This is an Open Access article distributed under the terms of the Creative Commons Attribution License (http://creativecommons.org/licenses/by/2.0, accessed on 7 August 2025), which permits unrestricted use, distribution, and reproduction in any medium, provided the original work is properly cited. *We strongly recommend reading this statement in conjunction with the CONSORT 2010 Explanation and Elaboration for important clarifications on all the items. If relevant, we also recommend reading CONSORT extensions for cluster randomised trials, non−inferiority and equivalence trials, non−pharmacological treatments, herbal interventions, and pragmatic trials. Additional extensions are forthcoming: for those and for up−to−date references relevant to this checklist, see https://www.consort-spirit.org/, accessed on 7 August 2025.

**Figure 4 ijms-26-09286-f004:**
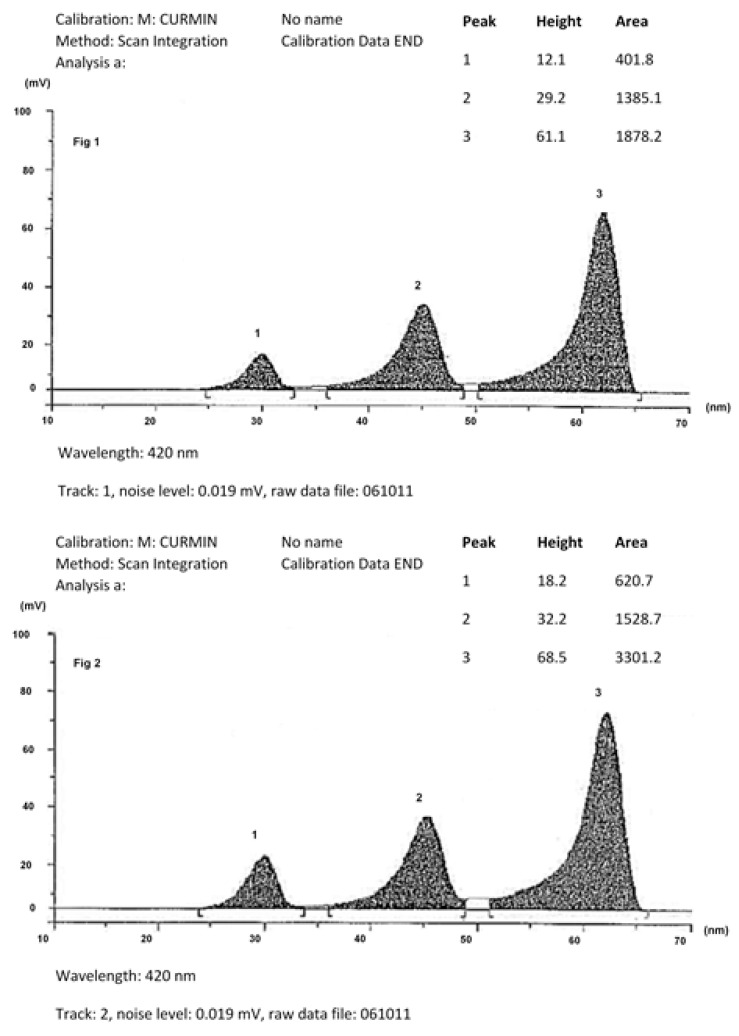
Chromatographic fingerprints of curcuminoids extract.

**Table 1 ijms-26-09286-t001:** Baseline characteristics of study participants.

Characteristics	Placebo Median (IQR) *n* = 114	Curcumin Median (IQR) *n* = 113	*p*-Values *
Sex (M/F ratio)	42/67 (0.63)	50/65 (0.77)	0.54 ^†^
Age (years)	63 (13)	61 (12)	0.07
BMI (kg/m^2^)	26.48 (4.90)	26.56 (5.58)	0.54
Waist circumference (cm)	93 (13)	92 (12)	0.29
TAC (mmol/L)	1.60 (0.17)	1.59 (0.185)	0.23
Glutathione peroxidase (U/L)	6534 (3152)	6447 (3454)	0.97
Superoxide dismutase (U/mL)	231 (62)	231 (68)	0.70
Malonaldehyde (μmol/L)	2.03 (0.63)	1.98 (0.64)	0.16
IL1-beta (pg/mL)	0.41 (0.35)	0.40 (0.445)	0.60
TNF-α (pg/mL)	5.28 (2.64)	4.40 (2.64)	0.17
Glucose (mg/dL)	120 (23)	121 (22)	0.67
HbA1c (%)	6.2 (0.7)	6.2 (0.75)	0.77
QUICKI	0.302 (0.02)	0.302 (0.03)	0.63
TC (mg/dL)	164 (40)	165 (38.5)	0.75
TG (mg/dL)	131 (88)	116 (73.5)	0.06
LDL (mg/dL)	99 (32)	97 (31.5)	0.88
NEFA (μmol/L)	0.80 (0.62)	0.90 (0.615)	0.21
TyG-WC	833.19 (153.43)	811.10 (138.31)	0.11
Liver stiffness (kPa)	5.4 (3.2)	5.4 (2.6)	0.65
CAP (dB/m)	230 (84)	222 (58.5)	0.70
History of cerebrovascular disease	7 (5.2%)	5 (3.7%)	0.30 ^†^
History of coronary artery disease	9 (6.7%)	8 (5.9%)	0.80 ^†^
History of hypertension	82 (61.2%)	76 (51.2%)	0.68 ^†^
History of dyslipidemia	104 (77.6%)	101 (74.8%)	0.84 ^†^

* Data were evaluated by the Mann-Whitney *U* test, except for sex (M:F ratio) and category variables. † *p* values were evaluated by Chi-square test. BMI = body mass index; CAP = controlled attenuation parameter; HbA1c = glycated hemoglobin; HDL = high-density lipoprotein; HOMA-IR = homeostatic model assessment of insulin resistance; hs-CRP = high-sensitivity C-reactive protein; IL-1β = Interleukin-1 beta; LDL = low-density lipoprotein; MDA = malondialdehyde; NEFA = non-esterified fatty acid; RANSEL = rapid assessment of nutritional status and insulin resistance; Ransod = randox superoxide dismutase assay; TAC = total antioxidant capacity; TBF = total body fat; TC = total cholesterol; TG = triglyceride; TNF-α = tumor necrosis factor-alpha; TyG-WC = triglyceride-glucose index-waist circumference; VF = visceral fat.

**Table 2 ijms-26-09286-t002:** Comparison of liver health between groups.

Outcomes	Follow Up Period (mo)	Placebo	Curcumin	*p*-Values *
CAP (dB/m)	0	230 (84)	227 (58.5)	NS
	3	231 (68)	225 (68.5)	** *0.01* **
	6	253 (72)	209 (71.5)	** *<0.001* **
	9	249 (85)	195 (70)	** *<0.001* **
	12	259 (74)	173 (78.5)	** *<0.001* **
Liver stiffness (kPa)	0	5.4 (3.2)	5.4 (2.6)	NS
	3	5.8 (2.2)	5.6 (2.3)	NS
	6	6.7 (2.2)	5.3 (2.3)	** *<0.001* **
	9	6.9 (3.6)	4.4 (1.4)	** *<0.001* **
	12	6.9 (2.8)	3.9 (1.5)	** *<0.001* **

* *p* values were evaluated by the Mann-Whitney *U* test. Significant *p* values were shown in bold and italic. CAP = controlled attenuation parameter.

**Table 3 ijms-26-09286-t003:** Comparison of oxidative stress and inflammatory markers between groups.

Outcomes	Follow Up Period (mo)	Placebo	Curcumin	*p*-Values *
TAC (μmol trolox eq/l)	0	1.60 (0.17)	1.59 (0.19)	NS
	3	1.72 (0.26)	1.75 (0.22)	** *0.040* **
	6	1.66 (0.19)	1.71 (0.22)	** *0.02* **
	9	1.70 (0.32)	1.80 (0.20)	** *0.006* **
	12	1.63 (0.23)	1.85 (0.16)	** *<0.001* **
GPx (U/mL)	0	6534 (3152)	6447 (3454)	NS
	3	6274 (2585)	7143 (3005)	** *0.01* **
	6	6748 (2472)	8000 (3016.5)	** *<0.001* **
	9	5467 (1639)	9912 (2261.5)	** *<0.001* **
	12	4837 (1159)	12,587 (3586)	** *<0.001* **
SOD (U/mL)	0	231 (62)	231 (68)	NS
	3	232 (67)	251 (45)	** *0.035* **
	6	210 (39)	268 (44.5)	** *<0.001* **
	9	206 (23)	278 (37.5)	** *<0.001* **
	12	178 (30)	315 (69.5)	** *<0.001* **
MDA (μmol/L)	0	2.03 (0.63)	1.98 (0.64)	NS
	3	1.90 (0.77)	2.09 (0.70)	** *0.03* **
	6	2.34 (0.59)	1.95 (0.81)	** *<0.001* **
	9	2.32 (0.74)	1.67 (0.68)	** *<0.001* **
	12	2.40 (0.80)	1.32 (0.56)	** *<0.001* **
IL1-beta (pg/mL)	0	0.41 (0.35)	0.40 (0.45)	NS
	3	0.40 (0.42)	0.49 (0.46)	NS
	6	0.89 (0.20)	0.52 (0.27)	** *<0.001* **
	9	0.92 (0.10)	0.42 (0.24)	** *<0.001* **
	12	0.93 (0.07)	0.31 (0.18)	** *<0.001* **
TNF-α (pg/mL)	0	5.28 (2.64)	4.40 (2.64)	NS
	3	5.28 (1.76)	5.28 (2.64)	NS
	6	5.65 (2.27)	4.01 (1.76)	** *<0.001* **
	9	6.69 (2.53)	3.98 (1.37)	** *<0.001* **
	12	6.99 (2.92)	3.29 (1.17)	** *<0.001* **

* *p* values were evaluated by the Mann-Whitney *U* test. Significant *P* values were shown in bold and italic. GPx = glutathione peroxidase; IL-1β = Interleukin-1 beta; MDA = malondialdehyde; SOD = superoxide dismutase assay; TAC = total antioxidant capacity; TNF-α = tumor necrosis factor-alpha.

**Table 4 ijms-26-09286-t004:** Comparison of metabolic parameters between groups.

Outcomes	Follow Up Period (mo)	Placebo	Curcumin	*p*-Values *
BMI (kg/m^2^)	0	26.48 (4.90)	26.56 (5.58)	NS
	3	26.90 (4.82)	25.40 (4.44)	** *0.036* **
	6	26.89 (4.71)	24.98 (3.96)	** *0.008* **
	9	27.48 (5.24)	25.00 (4.98)	** *<0.001* **
	12	28.69 (5.26)	24.90 (4.93)	** *<0.001* **
WC (cm)	0	93 (13)	92 (12)	NS
	3	93 (13)	90 (12.5)	** *0.04* **
	6	94 (11)	92 (11)	** *<0.001* **
	9	95 (14)	90 (11)	** *<0.001* **
	12	95 (14)	88 (11)	** *<0.001* **
TyG-WC	0	833.192 (153.428)	811.101 (138.313)	NS
	3	840.894 (161.246)	801.300 (140.731)	** *0.002* **
	6	891.349 (126.558)	818.757 (101.063)	** *<0.001* **
	9	853.660 (150.686)	783.073 (104.690)	** *<0.001* **
	12	865.939 (163.763)	777.063 (105.613)	** *<0.001* **
Glucose (mg/dl)	0	120 (23)	121 (22)	NS
	3	126 (26)	124 (22)	NS
	6	125 (27)	122 (27)	** *0.02* **
	9	129 (21)	117 (22.5)	** *<0.001* **
	12	133 (22)	114 (24)	** *<0.001* **
HbA1c (%)	0	6.2 (0.7)	6.2 (0.8)	NS
	3	6.4 (0.7)	6.2 (0.7)	NS
	6	6.4 (0.7)	6.1 (0.9)	** *0.01* **
	9	6.4 (0.7)	6.1 (0.9)	** *<0.001* **
	12	6.4 (0.9)	6.0 (0.8)	** *<0.001* **
QUICKI	0	0.302 (0.02)	0.302 (0.03)	NS
	3	0.299 (0.02)	0.305 (0.02)	** *0.04* **
	6	0.300 (0.02)	0.308 (0.02)	** *0.009* **
	9	0.296 (0.02)	0.304 (0.02)	** *<0.001* **
	12	0.294 (0.02)	0.306 (0.03)	** *<0.001* **
HOMA-IR	0	5.0 (2.5)	5.10 (3.35)	NS
	3	5.5 (2.8)	4.5 (2.85)	** *0.02* **
	6	5.5 (2.8)	4.4 (2.85)	** *0.003* **
	9	5.9 (2.6)	4.6 (2.90)	** *<0.001* **
	12	6.3 (2.6)	4.4 (2.60)	** *<0.001* **

* *p* values were evaluated by the Mann-Whitney *U* test. Significant *p* values were shown in bold and italic. BMI = body mass index; HbA1c = glycated hemoglobin; HOMA-IR = homeostatic model assessment of insulin resistance; QUICKI = quantitative insulin sensitivity check index; TyG-WC = triglyceride-glucose index-waist circumference; WC = waist circumference.

**Table 5 ijms-26-09286-t005:** Comparison of lipid profile between groups.

Outcomes	Follow Up Period (mo)	Placebo	Curcumin	*p*-Values *
TC (mg/dl)	0	164 (40)	165 (38.5)	NS
	3	169 (40)	163 (34)	NS
	6	179 (45)	161 (32.5)	** *<0.001* **
	9	181 (40)	160 (35.5)	** *<0.001* **
	12	184 (43)	159 (36)	** *<0.001* **
TG (mg/dl)	0	131 (88)	116 (73.5)	NS
	3	132 (94)	111 (69.5)	** *0.003* **
	6	148 (97)	110 (66)	** *<0.001* **
	9	139 (96)	107 (72.5)	** *<0.001* **
	12	132 (99)	107 (66.5)	** *<0.001* **
LDL (mg/dl)	0	99 (32)	97 (31.5)	NS
	3	93 (31)	91 (27)	NS
	6	110 (40)	98 (35)	** *0.03* **
	9	99 (36)	85 (28)	** *<0.001* **
	12	101 (37)	87 (27.5)	** *<0.001* **
NEFA (μmol/L)	0	0.80 (0.62)	0.90 (0.62)	NS
	3	1.04 (0.45)	1.15 (0.54)	NS
	6	0.82 (0.48)	1.16 (0.80)	** *<0.001* **
	9	1.10 (0.57)	0.86 (0.51)	** *<0.001* **
	12	1.20 (0.70)	0.76 (0.51)	** *<0.001* **

* *p* values were evaluated by the Mann-Whitney *U* test. Significant *p* values were shown in bold and italic. LDL = low-density lipoprotein; NEFA = non-esterified fatty acid; TC = total cholesterol; TG = triglyceride.

**Table 6 ijms-26-09286-t006:** Adverse effects in curcumin-treated group and placebo-treated group.

Adverse Effects	Placebo (*n* = 114) *	Curcumin (*n* = 113) *
Abdominal pain	-	13 (11.5)
Diarrhea	-	8 (7.1)
Headache	2 (1.7)	5 (4.4)

* Values expressed as number (percentage).

**Table 7 ijms-26-09286-t007:** Parameters and adverse effects in the curcumin-treated and placebo-treated groups at each follow-up visit.

Variables	Visit	Placebo		Curcumin		*p* Value
		Mean (SEM)	Min-Max	Mean (SEM)	Min-Max	
Creatinine (mg/dL)	Baseline	0.87 (0.02)	0.40–1.69	0.86 (0.02)	0.45–1.6	0.77
	3 mo	0.88(0.02)	0.45–1.81	0.91 (0.05)	0.46–7.26	0.64
	6 mo	0.94(0.02)	0.47–2.04	0.92 (0.02)	0.52–1.70	0.40
	9 mo	0.94 (0.02)	0.44–1.83	0.93 (0.02)	0.54–1.81	0.51
	12 mo	0.87(0.02)	0.40–1.69	0.85(0.02)	0.45–1.60	0.77
Aspartate aminotransferase (U/L)	Baseline	25.01 (0.87)	11–89	25.34 (0.80)	13–67	0.58
	3 mo	22.45 (0.79)	9–78	23.85(0.90)	11–111	0.076
	6 mo	23.53 (1.43)	8–214	24.12 (0.98)	10–89	0.47
	9 mo	21.78 (0.63)	11–76	24.29 (1.23)	12–114	0.88
	12 mo	25.01(0.87)	11–89	25.41(0.81)	13–67	0.54
Alanine aminotransferase (U/L)	Baseline	27.58 (1.56)	5–145	30.09 (1.50)	5–118	0.08
	3 mo	24.08 (1.1)	6–101	27.49 (1.7)	6–214	0.08
	6 mo	24.74 (1.15)	7–98	28.16 (1.64)	7–186	0.18
	9 mo	23.01 (1.14)	6–117	27.50 (1.80)	6–129	0.36
	12 mo	27.58(1.56)	5–145	30.27(1.51)	8–118	0.21

**Table 8 ijms-26-09286-t008:** Baseline use of antihypertensive and antidyslipidemic medications.

Medications	Placebo (*n* = 114)	Curcumin *(n* = 113)	*p*-Value ^†^
Antihypertensive Medications *			
Angiotensin receptor blockers	80 (70.2)	86 (76.1)	0.39
Calcium channel blockers	26 (22.8)	18 (15.9)	0.25
Beta blockers	21 (18.4)	17 (15.0)	0.61
Statins	59 (51.8)	55 (48.7)	0.74

* Values expressed as number (percentage); ^†^
*p*-values were determined by a Chi-squared test.

**Table 9 ijms-26-09286-t009:** Mean daily intake of nutrients at baseline and at 12 Months, by Group.

Daily Intake of Nutrients	Placebo (*n* = 114)		Curcumin (*n* = 113)		*p* Value ^2^
	Baseline ^1^	12 mo	Baseline ^1^	12 mo	
Energy (kcal/d)	1857.60 ± 110.97	1893.74 ± 74.30	1864.21 ± 87.98	1881.16 ± 67.23	0.100
Carbohydrate (% of energy)	57.50 ± 2.57	58.07 ± 2.56	57.04 ± 1.52	58.08 ± 1.62	0.148
Protein (% of energy)	12.98 ± 2.13	13.21 ± 1.35	13.33 ± 1.28	13.44 ± 1.20	0.418
FAT (% of energy)	28.27 ± 2.12	28.91 ± 1.91	28.47 ± 2.42	28.44 ± 2.23	0.056
Fiber (g/d)	8.54 ± 1.16	8.46 ± 0.88	8.49 ± 0.82	8.39 ± 0.64	0.464

Groups at baseline for any variable by *t* test. ^1^ All parameters are presented as means ± SDs. There are no significant differences between the two ^2^ The curcumin had no significant effect on mean daily intake of nutrients by one-factor ANCOVA with the baseline value as the covariate. There were no significant differences in the daily mean-energy (energy, carbohydrate, protein, fat, and fiber) and nutrient intakes between the curcumin and placebo groups.

**Table 10 ijms-26-09286-t010:** Capsule Consumption by Subjects Per Day and Per 3 Months, Counted at 3-, 6-, 9-, and 12-Month Visits.

	Visit	Placebo		Curcumin		*p* Value
		Number of Subjects Assessed	Number of Capsules Taken *	Number of Subjects Assessed	Number of Capsules Taken *	
Consumption per 3 Months	3 mo	114	522.90 (54.59)	113	514.80 (50.11)	0.48
	6 mo	114	524.7 (21.61)	113	512.10 (25.51)	0.23
	9 mo	114	517.16 (20.32)	113	514.96 (21.33)	0.43
	12 mo	114	515.06 (20.82)	113	514.11 (21.78)	0.33
Consumption per Day	3 mo	114	5.81 (0.60)	113	5.72 (0.58)	0.57
	6 mo	114	5.83 (0.23)	113	5.69 (0.28)	0.13
	9 mo	114	5.75 (0.22)	113	5.72 (0.24)	0.43
	12 mo	114	5.72 (0.21)	113	5.71 (0.25)	0.33

* The data are presented as the mean ± SEM.

## Data Availability

The main findings from this study are included in the article. For additional data, please contact the corresponding author.

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
