# Peer review of "Curcumin Attenuates Liver Steatosis via Antioxidant and Anti-Inflammatory Pathways in Obese Patients with Type 2 Diabetes Mellitus: A Randomized Controlled Trial"

_ijms, 2025, doi:10.3390/ijms26199286_

Round 1
Reviewer 1 Report
Comments and Suggestions for Authors
The article titled “Curcumin attenuates liver steatosis via antioxidant and anti-inflammatory pathways in obese patients with type 2 diabetes mellitus: A randomized controlled trial” presents a randomized, double-blind, placebo-controlled trial evaluating the effect of curcumin supplementation on liver steatosis and metabolic dysfunction in obese patients with type 2 diabetes mellitus (T2DM), and it demonstrates several strengths including a large sample size, extended 12-month intervention, comprehensive biochemical analyses, and rigorous trial design with clear control of confounders such as diet and antihyperglycemic therapy. The authors convincingly show that curcumin significantly improved multiple metabolic and hepatic parameters, including liver fat accumulation, insulin resistance markers, antioxidant activity, lipid metabolism, and inflammatory cytokines, supporting its therapeutic value in MASLD. However, while the outcomes are robust and consistent with prior biochemical evidence, some limitations need deeper acknowledgment. The reviewer has the following comments that the authors need to address:
- The absence of histological confirmation via liver biopsy limits the diagnostic reliability and precise classification of MASLD, leaving potential gaps in correlating imaging-based CAP reductions with definitive histopathological improvements.
- Excluding patients with poor glycemic control results in a highly selected study population, which may limit the generalizability of the findings to real-world MASLD–T2DM cohorts that frequently include individuals with moderate-to-severe metabolic dysregulation.
- The use of a single fixed dose limits the ability to evaluate dose–response relationships and identify optimal therapeutic windows, which are essential considerations for translational and clinical application.
- Some interpretations tend to overstate the clinical significance; although improvements in biochemical and imaging markers are noted, the lack of long-term clinical endpoints such as fibrosis regression, prevention of cirrhosis, or cardiovascular outcomes limits the strength of evidence for recommending curcumin in routine clinical practice.
- While safety conclusions are reassuring due to the absence of reported adverse events, they would be strengthened by more detailed information on patient adherence, gastrointestinal tolerance, and potential herb–drug interactions.
- The external validity of the findings is limited by the single-center Thai population; validation in broader, multi-ethnic cohorts is warranted to enhance generalizability.
- The manuscript could be further strengthened by citing recent review on diarylheptanoids, as these compounds are structurally related to curcumin and have demonstrated comparable antioxidant and anti-inflammatory properties. Including this context would enrich the discussion on structure–activity relationships and potential translational relevance for improving liver steatosis and metabolic inflammation.
https://aces.onlinelibrary.wiley.com/doi/full/10.1002/asia.202400380
Reviewer 2 Report
Comments and Suggestions for Authors
Title of manuscript: Curcumin Attenuates Liver Steatosis via Antioxidant and Anti-Inflammatory Pathways in Obese Patients with Type 2 Diabetes Mellitus: A Randomized Controlled Trial
A carefully planned randomized, double-blind, placebo-controlled study assessing the effects of curcumin supplementation in obese T2DM patients with MASLD is reported in this publication. With a lengthy follow-up (12 months) and a comparatively high sample size, the study tackles a significant clinical concern. The results are encouraging, but before the book is ready for publication, a few things need to be clarified and improved.
Major Comments:
Abstract
The abstract is informative but overly detailed. It should be more concise, focusing on the most clinically relevant results (e.g., CAP reduction, HbA1c change, key inflammatory markers).
Introduction
The introduction is well written but contains redundancy regarding MASLD–T2DM bidirectional relationship. It should more clearly highlight the specific knowledge gap that this trial addresses (e.g., lack of long-term RCTs with curcumin).
Methods
- The diagnosis of liver steatosis relied solely on CAP (FibroScan). The absence of liver biopsy or MRI-PDFF should be acknowledged more explicitly and justified.
- Restricting enrollment to patients only on metformin limits external validity. This should be addressed in the limitations section.
Results
- Table 5 is overloaded with data. Consider splitting it into multiple tables (e.g., metabolic outcomes, oxidative stress markers, inflammatory cytokines).
- Please clarify whether observed improvements are clinically meaningful or only statistically significant.
Discussion
- The discussion restates results rather than deeply interpreting mechanisms. Greater emphasis should be placed on how curcumin’s known mechanisms (antioxidant, anti-inflammatory, lipid-lowering) align with the trial’s findings.
- Lack of comparison with modern pharmacotherapies (GLP-1 receptor agonists, SGLT2 inhibitors) is a limitation that should be discussed.
Conclusion
The conclusion should be more cautious. Current evidence supports curcumin as a promising adjunctive therapy, not a standard treatment.
Minor Comments:
Abstract
Consider including the clinical trial registration number (TCTR ID) in the abstract for transparency.
Introduction
Some cited references are outdated (before 2015). Please add recent evidence (2022–2024) on curcumin and MASLD/T2DM.
Methods
- Capsule compliance was assessed by pill counts, but plasma curcumin levels were not measured. Consider discussing this limitation.
- Table 1 (antihypertensive/antidyslipidemic drugs) could be simplified to improve readability.
Results
- A graphical representation (line charts for CAP, HbA1c, BMI over time) would enhance clarity.
- The NEFA results section appears inconsistent: an increase at 3 months followed by a decrease later. This needs clearer explanation.
Discussion
- The safety discussion should mention potential absorption issues and interactions with other drugs.
- Some paragraphs are lengthy and repetitive; condensation would improve readability.
Conclusion
Suggest adding a statement about the need for multicenter studies with different doses and longer follow-up.
Overall Style / Formatting
- Abbreviations (e.g., GPx, SOD, MDA) should be defined at first mention and consistently used.
- The figures should be re-ordered to match the sequence in which they are cited in the Results section, ensuring clarity and consistency.
- Tables and figures should be reformatted for readability.
- The manuscript would benefit from professional language editing to remove redundancy.
Final Recommendation
The manuscript is scientifically sound and potentially publishable, but requires major revisions to improve clarity, interpretation, and generalizability.
Reviewer 3 Report
Comments and Suggestions for Authors
This study is a randomized, double-blind, placebo-controlled trial aimed at investigating the effects of curcumin on hepatic steatosis in obese patients with type 2 diabetes mellitus (T2DM) and its potential mechanisms. The study design is reasonable, with a relatively long follow-up period, providing certain clinical reference value. However, several issues remain to be addressed.
Specific concerns:
1.Line 36: The abstract mentions "no severe adverse reactions." It is recommended to clarify whether systematic evaluations were conducted on key metabolic organs (such as the liver and kidneys) and to supplement the Discussion section with documented non-serious adverse events (e.g., gastrointestinal discomfort) and their incidence rates, in order to comprehensively assess the long-term safety of curcumin.
2.Line 215: Please clarify why controlled attenuation parameter (CAP) was chosen over magnetic resonance imaging-derived proton density fat fraction (MRI-PDFF) for assessing hepatic steatosis. If MRI-PDFF was not adopted due to feasibility or cost constraints, the limitations of the CAP method and the potential risk of misclassification should be clearly stated in the discussion to help readers understand the results more accurately.
3.Line 333: It is claimed in section 3.1.3 that curcumin significantly reduced IL-6 levels; however, no IL-6 data are provided in Table 5 or the baseline characteristics table. It is recommended to verify and supplement the relevant data, or to revise the inconsistent statements in the text.
4.The discussion section needs to deepen the interpretation of the results, rather than simply reiterating them. For example, at Line 412, it should not just state "Curcumin reduced inflammatory factors (TNF-α, IL-1β)," but should further elaborate: "The improvement in hepatic steatosis occurred synchronously with a significant reduction in circulating inflammatory markers, indicating that anti-inflammatory effects are a core mechanism by which curcumin alleviates hepatic steatosis. Chronic low-grade inflammation is a key pathway connecting insulin resistance and MASLD, and curcumin may act by interrupting this vicious cycle."
Round 2
Reviewer 1 Report
Comments and Suggestions for Authors
The authors have thoroughly addressed the comments raised by the previous reviewers, demonstrating careful attention to detail and a commitment to improving the manuscript. In my opinion, the article, in its current form, meets the standards of quality and scientific rigor expected for publication.